# Pharmacological Justification for the Medicinal Use of *Plumeria rubra* Linn. in Cardiovascular Disorders

**DOI:** 10.3390/molecules27010251

**Published:** 2021-12-31

**Authors:** Imran Ahmad Khan, Musaddique Hussain, Shahzada Khurram Syed, Malik Saadullah, Ali M. Alqahtani, Taha Alqahtani, Afaf A. Aldahish, Saeed Asiri, Ling-Hui Zeng

**Affiliations:** 1Department of Pharmacology, The Islamia University of Bahawalpur, Bahwalpur 63100, Pakistan; 2Department of Basic Medical Sciences, School of Health Sciences, University of Management and Technology, Lahore 54000, Pakistan; shahzada.khurram@umt.edu.pk; 3Department of Pharmaceutical Chemistry, Government College University, Faisalabad 38000, Pakistan; maliksaadullah@gcuf.edu.pk; 4Department of Pharmacology, College of Pharmacy, King Khalid University, Abha 62529, Saudi Arabia; amsfr@kku.edu.sa (A.M.A.); ttaha@kku.edu.sa (T.A.); adahesh@kku.edu.sa (A.A.A.); 5Department of Clinical Laboratory Sciences, College of Applied Medical Sciences, Najran University, Najran 66218, Saudi Arabia; saaasiri@nu.edu.sa; 6Department of Pharmacology, Zhejiang University City College, Hangzhou 310015, China; zenglh@zucc.edu.cn

**Keywords:** *Plumeria rbra*, cardioprotective, adrenaline, troponin, CRP

## Abstract

*Plumeria rubra* (L.) is a traditional folkloric medicinal herb used to treat cardiovascular disorders. The present investigation was methodically planned to investigate the pharmacological foundations for the therapeutic effectiveness of *P. rubra* in cardiovascular illnesses and its underlying mechanisms. Ex vivo vaso-relaxant effects of crude leaf extract of *P. rubra* were observed in rabbit aorta ring preparations. Hypotensive effects were measured using pressure and force transducers connected to the Power Lab data acquisition system. Furthermore, *P. rubra* displayed cardioprotective properties in rabbits when they were exposed to adrenaline-induced myocardial infarction. In comparison to the intoxicated group, the myocardial infarction model showed decreased troponin levels, CK-MB, LDH, ALT, ALP, AST, and CRP, as well as necrosis, apoptosis, oedema, and inflammatory cell enrollment. *P. rubra* has revealed good antioxidant properties and prolonged the noradrenaline intoxicated platelet adhesion. Its anticoagulant, vasorelaxant, and cardioprotective effects in both in vivo and ex vivo investigations are enabled by blocking L-type calcium channels, lowering adrenaline, induced oxidative stress, and tissue tear, justifying its therapeutic utility in cardiovascular disorders.

## 1. Introduction

Cardiovascular disease (CVD) is defined as a set of heart and blood vessel abnormalities that comprise congestive heart failure (CHF), myocardial infarction (MI), peripheral arterial disease (PAD), angina of different types, and coronary heart disease (CHD) [1]. Globally, hypertension is the primary cause of death in humans [2]. Many developing countries, such as India, Iran, Bangladesh, Afghanistan, and Pakistan, are fast catching up with this epidemic, due to drastically changing lifestyle habits [2]. Fast foods, additives, preservatives, biologically engineered food, and the sedentary lifestyle multiply the misery [3]. Although synthetic drugs are incredibly effective in treating cardiovascular diseases, their usage is restricted due to their side effects [4]. MI is a primary ischemic condition, characterized by the severe destruction of myocardial tissue. It occurs due to a mismatch between oxygen demand and blood supply in the myocardium, without atherosclerotic plaque or with atherosclerotic plaque [5]. Exaggerated production of reactive oxygen species (ROS) is accountable for lipid peroxidation of the endocardium. This damage results in the deletion of cardiac antioxidants, the elevation of oxidative stress, and apoptosis [5]. Similarly, catecholamine affects the myocardium in both inotropic and chronotropic ways. Surplus catecholamines cause coronary vasoconstriction, which increases myocardial oxygen demand and decreases myocardial blood delivery, resulting in MI [6,7]. Adrenaline (ADR) is a catecholamine produced by the adrenal glands’ medullary potion. ADR is an approved drug for cardiopulmonary resuscitation in humans. It also has therapeutic uses in treating allergic reactions, glaucoma, asthma, and cardiac arrest [8]. It causes MI above the therapeutic dose (2 mg/kg body weight) [5]. This method was used to assess the cardioprotective effect of test medications to induce MI in laboratory animals. Lipid peroxidation (LPO), which results in the loss of intracellular antioxidants, causes MI [6]. It leads to an increased production of nitrosative derivatives, which cause the overproduction of ROS and, as a result, the myocardium faces immense oxidative stress [9]. It increases Ca^++^ upsurge by opening L-type calcium channels on cardiomyocytes, and raises oxidative stress by escalating workload [1]. It causes aortic and coronary vasoconstriction at higher doses, and enhances blood coagulation by increasing clotting factor VIII, fibrinolysis, and platelet count [5].

Herbal remedies are highly beneficial in treating a variety of ailments all over the world; herbal medicines continue to provide vital curative agents in conventional and modern medicine, and they are also less toxic than synthetic drugs [10].

*Plumeria rubra*, Linn. (Family: Apocynaceae) is native to Mexico, Central America, Colombia, and Venezuela but cultivated in tropical and subtropical countries. In South Asian native languages, it is known as ‘Lal Champa’, and Frangipani in English [11]. It is regarded as a great ornamental plant, and is frequently seen in graveyards, gardens, and parks. It is extensively cultivated in tropical and subtropical regions all over the world. Diarrhoea and emesis have traditionally been treated using a decoction of *P. rubra* bark and roots [12]. In Mexican traditional medicine, the latex and flower of *P. rubra* are used to treat tooth and earache [11]. High-performance liquid chromatography (HPLC) analysis revealed many vital phytoconstituents, such as α-allamcidin, α-amyrin, β-allamcidin, β-amyrin acetate, β-sitosterol, 13-*O*-p-Coumaroylplumieride, 15-Demethylplumieride, 2,4,6-trimethoxyaniline, 2,5-dimethoxy-p-benzoquinone, 3-*O*-caffeoylquinic acid, allamandin, allamcin, arjunolic acid, benzyl salicylate, betulinic acid, citric acid, fulvoplumierin, gaertneroside, isoplumericin, kaempferol, kaempferol-3-rutinoside, kaempferol-3-*O*-glucoside, liriodendrin, lupeol, lupeol acetate, lupeol carboxylic acid, maslinic acid, methyl salicylate, naphthalene, narcissin, nerolidol, oleanolic acid, oleic acid, P-(*E*)-coumaric acid, plumericidine, plumericin, plumerinine, plumerubroside, plumieride, plumieride-*E*-p-coumarate, quercetin 3-*O*-α-L-arabinopyranoside, quercitrin, quercetin, quinic acid, rubradoid, rubrajaleelol, rubrajaleelic acid, rubranonoside, rutin, scopoletin, stigmasterol, stigmast-7-enol, sweroside, taraxasteryl acetate, and ursolic acid [13]. The decoction of the bark has been used to cure venereal disease, as well as rheumatism, leprosy and fever in the indigenous system of medicine [11]. The root decoction treats bacterial infections, rheumatic pains, and cancers [12]. Pharmacological investigations reported analgesic, antidiabetic, antipyretic, anti-obesity, antimicrobial, lipid-lowering, anticancer, antioxidant, insecticidal, and gastroprotective effects [14]. It was written to treat cardiovascular ailments in folk medicines [15]. However, there is no recorded pharmacological validation about its utility in cardiovascular disorder. The current investigation evaluated the anticoagulant, hypotensive and cardioprotective effect of the aqueous-methanolic leaf extract of P. rubra in rabbits, using in vivo, in vitro and ex vivo experiments. The rabbit animal model was chosen over rats because rabbits have a more similar profile to humans. The calcium channels of rabbits’ aorta are more diversified than that of rats [6]. 

## 2. Results

### 2.1. Phytochemical Evaluation

Phytochemical investigation validated the presence and high presence of flavonoids, phenols, tannins, anthraquinones, saponins, coumarins, and alkaloids in the aqueous-methanolic leaf extract of *P. rubra* by visually observing the prescribed colour change and precipitate formation of the extract (Table 1).

### 2.2. HPLC Analysis

HPLC analysis validated many phytoconstituents in varying concentrations, among them the essential phytochemicals rutin, isoquercetin, kaempferol, and plumericin were identified based on retention time (Figure 1).

### 2.3. DPPH Assay

Aqueous-methanolic leaf extract of *P. rubra* showed good antioxidant potential with the inhibitory concentration of 138 µg/mL about ascorbic acid in DPPH assay (Figure 2).

### 2.4. Evaluation of Myocardial Infarction 

ADR considerably elevates the level of cardiac biomarkers (troponin, CK-MB, LDH) and cardiohepatic biomarkers (ALT, CRP, ALP, AST) (*p* < 0.05) regarding control. The rabbits in the ADR-intoxicated group (group 2) had significantly higher cardiac markers, whereas the three groups treated with *P. rubra* at different doses (100, 200, and 300 mg/kg body weight) demonstrated a dose-dependent resistance to the ADR-intoxicated cardiac damage. In contrast, the three groups receiving *P. rubra* leaf extract had lower average levels of troponin, CK-MB, LDH and AST, ALT, CRP, ALP (*p* < 0.05) than the ADR-intoxicated group (Figure 3 and Figure 4).

#### Effect on Heart to Bodyweight Ratio

ADR noticeably increased the ratios regarding the control group. All three groups treated with *P. rubra* reduced their heart to bodyweight ratios less than the ADR-intoxicated group (Figure 5).

### 2.5. Histopathology 

Examination of heart tissue fragments from the ADR-intoxicated group revealed a notable variation in cardiac cell architecture. Histological changes in the ADR-intoxicated group included interstitial oedema, the shred of muscular fibers, cellular infiltration, vacuolar disintegration, disintegration, mottled staining, capillary enlargement, hemorrhage, and myocardium obstruction. All of the groups had brutal necrotic lesions, but the *P. rubra*-treated groups had less myocardial degradation (Figure 6).

### 2.6. Aortic Tissue Preparation and Vasorelaxant Activity

The isolated rabbit aorta rings showed a continuous contractile activity when exposed to phenylephrine (PE) (1 M) and K^+^ (80 mM). *P. rubra* elicited concentration-dependent relaxation of PE-induced spastic contractions (EC_50_ = 0.67 mg/mL; Cl 95%: 0.3849–1.188). Likewise, *P. rubra* elicited concentration-dependent relaxation in the aorta ring preparation against K^+^ (80 mM)-induced spastic contractions (EC_50_ = 1.30 mg/mL; Cl 95%: 0.9142–1.848). Similarly, verapamilat at a concentration of 1–3 μM, relaxed the PE (1 M) and K+ (80 mM)-induced contractions having EC_50_ = 0.11 μM (95% CI: 0.03–0.18 μM; *n* = 5) and EC_50_ = 0.03 μM (95% CI: 0.01–0.16 μM; *n* = 5), respectively (Figure 7a–d).

### 2.7. Antiplatelet Aggregatory Effect

The addition of ADR (2 µM) to a suspension of rinsed human platelets resulted in a significant decrease in optical density at 600 nm, indicating platelet aggregation. At 37 °C, the aggregatory effect was observed. *P. rubra* (100, 200, and 300 µg/mL) reduced platelet aggregation in a dose-dependent pattern (* *p* < 0.05, ** *p* < 0.001, *** *p*< 0.0001) (Figure 8).

### 2.8. Acute Oral Toxicity Dose Test

In the acute oral toxicity test, *P. rubra* was found safe up to 3000 mg/kg body weight. No death or morbidity was recorded at any dose, in any animal exposed to *P. rubra* aqueous-methanolic leaf extract.

## 3. Discussion

CVDs, particularly ischemic heart disease, MI, and cardiac hypertrophy, are the leading causes of death and morbidity globally [2]. Plants are widely used to cure minor to fatal cardiovascular diseases, and the present investigation was planned in order to screen a plant preparation (*P. rubra*) which is widely used to treat cardiovascular ailments in Pakistan, without sound pharmacological-documented evidence. Generally, flowers and leaves possess different phytochemical constituents; most of the research on the biological activities of *P. rubra*’s essential oils and flowers is available [14]. Since ancient times, numerous medicinal plant treatments have been utilized to treat circulatory diseases. However, in terms of cellular and molecular approaches, no scientific evidence has been examined and published on the molecular mechanism of the cardioprotective potential of *P. rubra. P. rubra,* the medicinal plant in this article which has appeared to show pharmacotherapeutic potential in vitro, in vivo, and ex vivo in animal studies, and therefore may influence cardiovascular ailments. This indigenous medicinal herb has a preventive, therapeutic impact by blocking, modifying, and regulating the production of specific regulatory proteins such as contractile, structural, and glycoproteins. These proteins are involved in calcium levels and promote the function of mitochondria. The cardioprotective action of medicinal plants has been established by decreasing the damage in cardiomyocytes, endothelial cells, vascular smooth muscle cells, and monocytes [16]. The cardioprotective action in cardiomyocytes has been demonstrated by the opening of the KATP channel, increased secretion of atrial natriuretic peptides, apoptosis, inflammation suppression, cardiac hypertrophy, oxidative stress, and endothelial nitric oxide synthase–nitric oxide (NOS–NO) signaling pathways, have all been found to be positive effects of the medicinal plant [17]. An HPLC study of the aqueous-methanolic leaf extract of *P. rubra* confirmed the presence of rutin, isoquercitrin, kaempferol, and plumerciin (Figure 1). Rutin is well documented for the cellular and molecular remodeling of cardiovascular pathogenesis by inhibiting the ERK1/2 pathway [18], and isoquercitrin attenuates cardiac dysfunction via the AMPKα-dependent pathway [19]. Similarly, kaempferol is reported for cardioprotection via downregulating the ASK1/MAPK signaling pathways [20].

ADR is a non-selective adrenoreceptor agonist, well-reported for the development of AMI and LVH in experimental animals at various doses [7]. Studies have verified that ADR at lower doses increases the weight of the heart (LVH) due to oxidative stress, via an increase in blood pressure through positive ionotropic and chronotropic responses [8]. In comparison, ADR at much higher doses for two consecutive days causes cardiac oedema, apoptosis, necrosis, and endocardial damage resulting in AMI [8]. ADR noticeably elevated the level of cardiac biomarkers, troponin, LDH, CK-MB, and cardiohepatic biomarkers, ALP, AST, CRP, ALT (*p* < 0.001). The three groups receiving the *P. rubra* leaf extract exhibited that the average amount of cardiac and cardiohepatic biomarkers was less in comparison with the ADR-intoxicated group, which advocates its cardioprotective character (Figure 3 and Figure 4). 

The cardiac weight to body weight ratio regarding the ADR-intoxicated group was significantly less in all three groups treated with *P. rubra* (Figure 5), which indicated protection against LVH in a dose-dependent manner. A high quantity of flavonoids in plants was reported to be responsible for their cardioprotective effect [21,22], and *P. rubra* leaf extract was observed to be affluent in flavonoid content (Table 1), [13]. As a result, it stands to reason that the cardioprotective effect of *P. rubra* could be influenced by the presence of these flavonoids. Reduced CRP levels were recorded in all three groups treated with *P. rubra* aqueous-methanolic leaf extract, in comparison with the ADR-intoxicated group; that may be a result of the ascorbic acid or flavonoid-dependent antioxidant potential of *P. rubra* (Figure 2). Our study endorses the previous antioxidant studies of *P. rubra* [23,24]; saponins have been reported to lower lipid peroxidation levels dose-dependently [25]. The outcome of this investigation further strengthens the claim of already published studies on saponins or saponin-containing plant extracts’ cardioprotective effects in animals [26,27]. *P. rubra* was found to be rich in saponins (Table 1).

Cardiac weight to body weight ratio regarding the ADR-intoxicated group was significantly less in all three groups treated with *P. rubra* (Figure 5), which indicated protection against LVH in a dose-dependent manner. A high quantity of flavonoids in plants was reported to be responsible for their cardioprotective effect [21,22], and *P. rubra* leaf extract was observed to be affluent in flavonoid content (Table 1), [13]. As a result, it stands to reason that the early research [28] found that the aqueous-methanolic leaf extract of *P. rubra* had a significant LPO inhibitory effect, which supports its potential as a cardioprotective agent, because LPO is a main contributory factor in ADR-intoxicated MI [1,5,6]. 

Histopathology of the ventricular portion of the ADR-intoxicated group revealed tremendous damage in cardiac cellular architecture, such as the disintegration and tearing of muscular fibers, capillary distention, vacuolar disintegration, mottled staining, interstitial oedema, mononucleate cellular infiltration, hemorrhage, and myocardium obstruction (Figure 6a). Less inflammatory cells and myocardial deterioration were observed in all three groups treated with *P. rubra*, while ruthless necrotic lesions were observed in the ADR-intoxicated group (Figure 6), which endorses the results of the biochemical investigation (Figure 3 and Figure 4), and justifies its use as a cardioprotective agent. 

ADR increases the Ca^++^ influx by opening the L-type voltage-gated calcium channels located on the myocardium, or increasing the release from intracellular stores of cardiomyocytes [5,7]. In conclusion, this exaggerated Ca^++^ increases the workload, resulting in oxidative stress [29]. The higher dose of ADR (2 mg/kg body weight) causes constriction of the aorta and coronary [7]. PE (1 μM) and K^+^ (80 mM) cause hypertension because of vasoconstriction, predominately by activating the alpha 1-adrenergic receptor and/or opening the L-type calcium channels [30], further building up oxidative stress (Figure 7). The vasodilator effect of *P. rubra* may depend on blocking these voltage-gated L-type calcium channels (Figure 7), as vasorelaxation was observed at a lesser concentration against K^+^ (80 mM)-induced contraction than PE (1 μM)-induced contraction. The compounds relaxed by the K^+^ (80 mM)-caused contraction are believed to be calcium channel blockers [1,5,6]. The negative ionotropic and chronotropic response was evident during the ex vivo investigation, similar to verapamil (Figure 7). Flavonoid-dependent vasodilatation was reported in the plants in numerous previous studies [21,22]. High flavonoid content was detected in aqueous-methanolic leaf extract of *P. rubra*, during the phytochemical investigation (Table 1, Figure 1), which provides logical grounds to believe its cardioprotective effect. Anticoagulant and antithrombotic medicines are integral parts of prescriptions of cardiovascular patients considered vital sources of cardioprotection in present-day pharmacology [31], whereas ADR is a well-documented procoagulant [7]. *P. rubra* noticeably decreased the platelet aggregatory effect of ADR (Figure 8), which provides another chapter on its use in cardiovascular disorders. 

## 4. Materials and Methods

### 4.1. Plant Materials

*P. rubra* (leaves) were collected fresh from the botanical garden of the Muhammad Institute of Medical and Allied Sciences, Multan. An expert taxonomist validated it at the IPAB, BZ, University, Multan. For future reference, the voucher specimen (P.Fl.565-1) has been deposited.

### 4.2. Extract Preparation

The fresh leaves were dried in the shade, and the vegetative waste/adulterants were separated by handpicking. The dried leaves were converted to coarse powder with a special herbal grinder. *P. rubra* (250 g) leaf powder was set for soaking in an aqueous-methanolic solvent (70:30 *v*/*v*) for nine days, in air-tight laboratory jars. The socked material was filtered through muslin cloth and Whatman-1 filter paper and with the help of a rotary evaporator. This filtrate was evaporated at 37 °C under reduced pressure [32]. The percent yield of the aqueous-methanolic leaf extract was calculated using this formula;
% age yield = Theoretical yield (g)/Actual yield (g) × 100(1)

### 4.3. Animals

Male albino rabbits of local bread, with an average weight of 1.5 kg, were acquired from the animal house of the Department of Pharmacology, the Islamia University of Bahawalpur. Animals were fed on standard food and tap water ad libitum. The temperature was maintained at 25 °C. All the experiments were carried out following NIH’s guidelines [33], and were approved by the Department of Pharmacology’s concerned committee (AS and RB/10/8/20).

### 4.4. Chemicals

Methanol was purchased from Lahore Laboratory Chemicals, Lahore, PB, Pakistan. Verapamil and phenylephrine were purchased from Abbott Laboratories, KPK, Pakistan (PVT) Ltd. Adrenaline was purchased from Ameer Pharmaceuticals, Lahore, PB, Pakistan. LDH, troponin, CK-MB and ALT, ALP, CRP, AST kits were obtained from Pakistan Scientific traders, Lahore, PB, Pakistan. All other chemicals/reagents and solutions utilized in the experiments were of analytical grade.

### 4.5. Preliminary Phytochemical Evaluation

Phytochemical screening was carried out to screen various phytochemical classes; glycosides, anthraquinones, alkaloids, tannins, flavonoids, and saponins in the *P. rubra* aqueous-methanolic leaf extract by using the protocol described earlier [34].

### 4.6. HPLC Analysis

HPLC was used to estimate phenolic acids in aqueous-methanolic leaf extract of *P. rubra* [13,35]. A binary gradient solvent system was used in HPLC, paired with a C-18 column with dimensions (250 × 4.6 mm), capable of separating 8–9 phenolics in 36 min at a flow-rate speed of 0.0008 µL/min, and a film thickness of 5 µm, with an oven set at 30 °C. The replicability for the separation of components was good with (run-to-run), rutin, kaempferol, plumericin, isoplumericin, and quercetin were prepared as reference (purity > 99 percent), obtained from Aldrich (St. Louis, MO, USA), and the dilutions were prepared with methanol to achieve 50 µg/mL. Samples were distinguished by comparing the *P. rubra* sample retention times to standards. The separation factor and resolution were used to evaluate the efficiency of separated components using HPLC, as shown in Figure 1.

### 4.7. Acute Oral Toxicity Dose Test

In 15 rabbits, the acute oral dose toxicity of *P. rubra* was tested. They were divided into 3 groups of 5 rabbits each; they were fasted for 24 h before receiving doses of 1000, 2000, and 3000 mg/kg orally. The rabbits were observed for 14 days after dosage for jerkiness, exhaustion, and mortality [36].

### 4.8. Determination of DPPH Assay

DPPH assay was carried out, as outlined in our previous study [5]. To summarize, different amounts of aqueous-methanolic leaf extract of *P. rubra* (4 mL) were appended to DPPH solution and prepared with up to 5 mL of methanol, then incubated in the dark for 40 min. At 517 nm, the absorbance of the incubated solution was measured using a spectrophotometer. All of the experiments were performed three times, and the percent inhibition was calculated in vitamin C equivalents. The DPPH scavenging outcome is calculated as:1% = A (blank) − B (sample)/A (blank) × 100(2)

### 4.9. Acute Myocardial Infarction Study

The animals were split into 5 groups, with 6 animals in each. The rabbits in Group 1 were given normal saline, whereas those in Group 2 were given ADR (2 mg/kg body weight subcutaneously) every 24 h, with a gap for 2 successive days. Group-3 rabbits were pre-treated with 100 mg/kg extract for 14 days in a row, before receiving ADR 2 mg/kg on the 14th and 15th days, staggered by 24 h. The rabbits of Group 4 were pre-treated with 200 mg/kg extract for fourteen days in a row and then given ADR 2 mg/kg SC. Every 24 h on the 14th and 15th days. Group-5 rabbits were pre-treated with extract at 300 mg/kg for fourteen days, and then ADR 2 mg/kg was administered SC at 24-h intervals on the 14th and 15th days, while *P. rubra* dosages were given by oral gavages [33]. The 3 doses were chosen based on previously published research on the *P. rubra’*s hepatoprotective property [36]. Rabbits were anaesthetized on the 16th day. Blood samples were collected from the rabbits’ marginal ear veins to analyze biochemical markers such as troponin, CK-MB, LDH and ALP, ALT, AST, and CRP levels in serum, measured using standard kits. 

#### Screening of Cardiac Weight to Bodyweight Ratio

Cardiac weight to bodyweight ratio helps key out cardiac weight index and necroses [1,6].

### 4.10. Histopathology

Rabbits were sacrificed under general anaesthesia, the heart was removed for histological analysis, and the ventricular component of the heart was quickly transferred to a 10% formalin solution. After that, the tissue was immersed in paraffin. A 5 μm segment was cut and stained with hematoxylin-eosin dye, before being mounted in xylene. [37]. Microscopic observation of the ventricular region of the cardiac tissue from different groups was utilized to evaluate the ADR effect alone, and in combination with the three groups of *P. rubra* on the cardiac cell structure. A compound microscope attached with a camera LCD was used to take micro-images. 

### 4.11. Isolated Aortic Tissue Preparation and Vasorelaxant Activity

The thorax cavity was excavated after the rabbits were slaughtered, and the aorta was carefully dissected. In a Petri dish containing Krebs’ solution, the fat and connective tissues were carefully removed from the aorta. Aortic rings (3–4 mm in length) were cut and placed in a tissue organ bath (10 mL) that already contained the Krebs–Henseleit solution, and was constantly bubbled with carbenogen gas (95 percent O_2_ and 5 percent CO_2_). The bottom hook was fixed, while the other was coupled to a force-displacement transducer connected to a Power Lab data collecting system, to record the isometric contractions. During the 40 min stabilization period at 1.0 g resting tension, Kreb’s solution was replaced every 15 min to prevent the accumulation of metabolites. After equilibration, the rings were pre-constructed with 1 × 10^−6^ M phenylephrine (PE) until the steady contractile curve (5–8 min) was reached, and the vasodilator effect of *P. rubra* was measured using a cumulative dosing method [38,39].

### 4.12. Calcium Channel Blocking Activity

*P. rubra* was tested against sustained spastic contractions caused by high K^+^ (80 mM) in the isolated rabbit aorta, which were conciliated by the opening of voltage-dependent Ca^++^ channels, and produced a contractile response by originating extracellular Ca^++^ influx. Chemicals that block Ca^++^ ions influx through these channels can reduce high K^+^ caused contractions. The administration of test material in an additive way against the persistent contractions is required to achieve a concentration-dependent inhibitory response [40]. The relaxant effect of the test drug against the simulated contractions is shown as a percentage (%) of the control contraction response. 

### 4.13. Adrenaline-Induced Platelet Activation and Aggregation

The blood sample was taken from the rabbit’s marginal ear vein and centrifuged (3000 rpm for 15 min) to obtain platelet-rich plasma. ADR (2 µM) was added to the samples as a supplement. Platelet aggregation was investigated using impedance aggregometry, whereas platelet activation was analyzed using flow cytometry before and after supplementation [41]. The effect of the 3 different administered doses of ADR on platelet aggregation (*n* = 5) was first analyzed. Platelet aggregation was next assessed after administering ADR in conjunction with *P. rubra* at doses of 100, 200, and 300 µg/mL (*n* = 5).

### 4.14. Statistical Analysis

The in vitro results for vasorelaxant activities were expressed as the mean ± SEM. EC50 values (with 95% CI) were calculated using the software Graph Pad Prism version-8 (Graph Pad Software, San Diego, CA, USA), and the dose–response curves were analyzed by the nonlinear regression sigmoidal response curve (variable slope). Similarly, in vivo results were evaluated by one-way analysis of variance (ANOVA) followed by Dunnett’s multiple comparison test. * *p* < 0.05, ** *p* < 0.001, *** *p* = 0.0001, was considered to be statistically significant.

## 5. Conclusions

*Plumeria rubra* was observed to produce vasorelaxant, hypotensive, anticoagulant, antioxidant, and calcium-channel-blocking effects. The cardioprotective effect of the *P. rubra* aqueous-methanolic leaf extract may be due to its diversity of phytoconstituents. Treatment with *P. rubra* may replenish antioxidants in cardiomyocytes, which are needed for resistance to oxidative damage generated by ADR. However, the actual molecular mechanism of cardioprotection is yet to be discovered. For further characterization, an LC-MS/MS spectrum is recommended. Furthermore, tannins, flavonoids, and cardiac glycosides were detected during the phytochemical screening, which may impact cardiovascular illnesses, particularly hypertension-induced LVH and MI. In closing, the therapeutic potential of *P. rubra* in cardiovascular disorders has been demonstrated in both ex vivo and in vivo studies; this will pave the way for new drug development to treat cardiovascular ailments modulating multiple pathways.

## Figures and Tables

**Figure 1 molecules-27-00251-f001:**
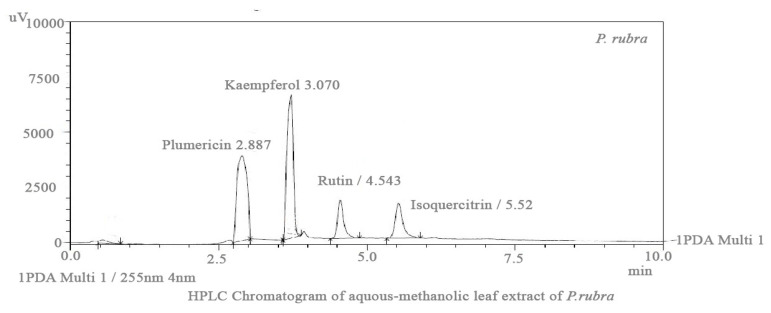
HPLC chromatogram of aqueous-methanolic leaf extract of *P. rubra* showing the rutin, isoquercetin, kaempferol and plumericin with reference to retention time of standard.

**Figure 2 molecules-27-00251-f002:**
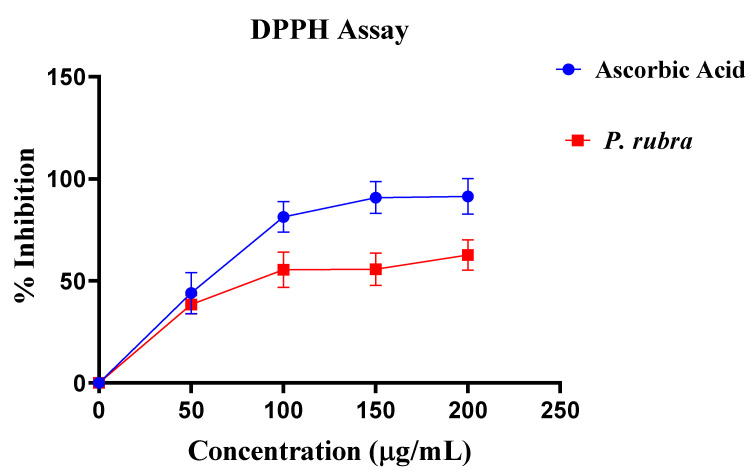
Antioxidant potential of *P. rubra* with respect to ascorbic acid by DPPH assay (*n* = 5).

**Figure 3 molecules-27-00251-f003:**
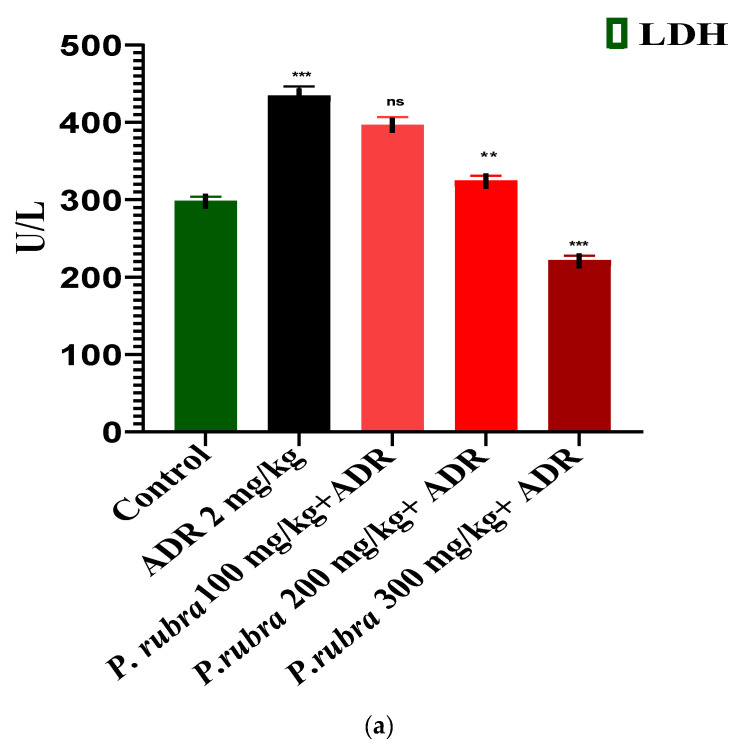
Cardioprotective effect of three doses of aqueous-methanolic leaf extract of *P. rubra* on the cardiac biomarkers, LDH (**a**), CK-MB (**b**), and troponin (**c**) against ADR-intoxicated MI. One-way ANOVA and Dunnett’s multiple comparison test were performed (** *p* < 0.005 & *** *p* < 0.001 & ns *p* > 0.005, *n* = 5).

**Figure 4 molecules-27-00251-f004:**
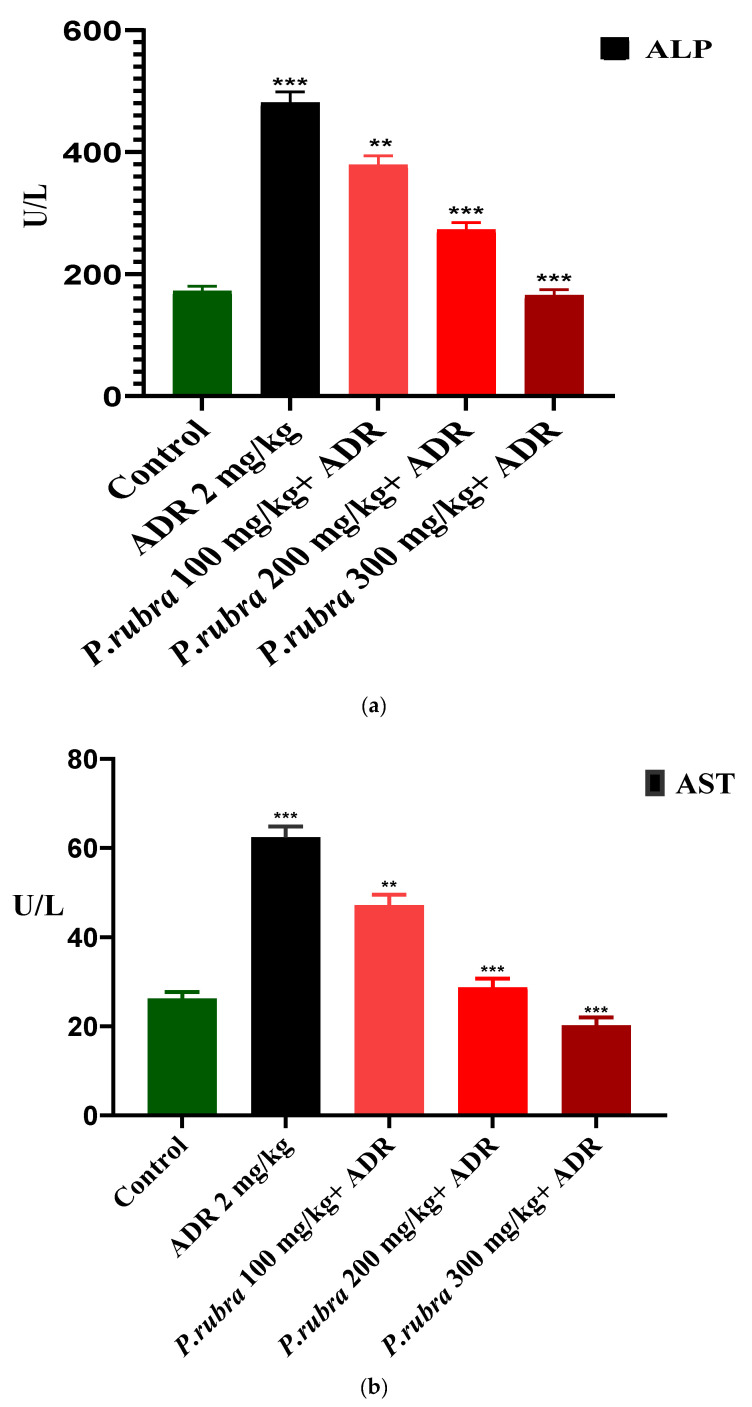
Cardioprotective effect of three doses of aqueous-methanolic leaf extract of *P. rubra* on the cardiohepatic biomarkers, ALP (**a**), AST (**b**), ALT (**c**) and CRP (**d**), against ADR-induced MI. One-way ANOVA and Dunnett’s multiple comparison test were performed (** *p* < 0.005, *** *p* < 0.001 & ns *p* > 0.005, *n* = 5).

**Figure 5 molecules-27-00251-f005:**
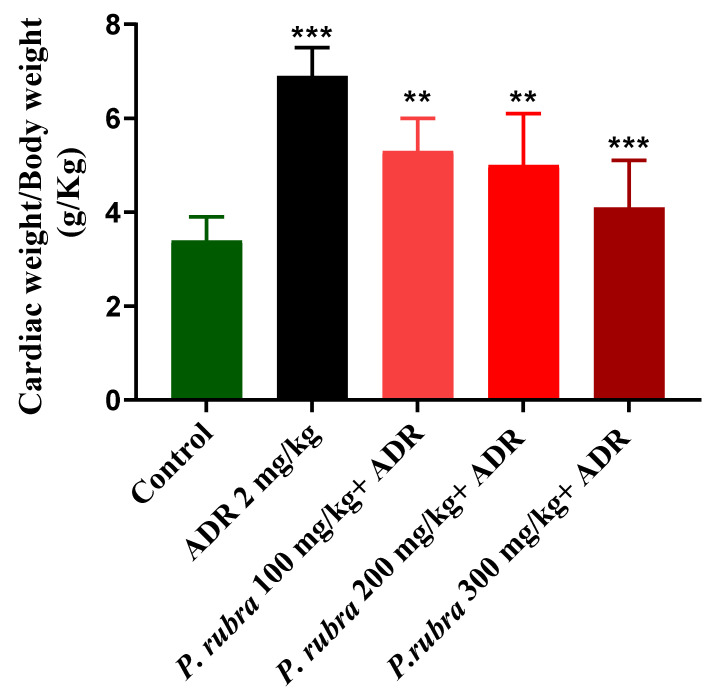
Cardioprotective effect of three doses of aqueous-methanolic leaf extract of *P. rubra* on heart weight to bodyweight ratio in ADR-intoxicated left ventricular hypertrophy and control. One-way ANOVA and Dunnett’s multiple comparison test were performed (** *p* < 0.005 & *** *p* < 0.001, *n* = 5).

**Figure 6 molecules-27-00251-f006:**
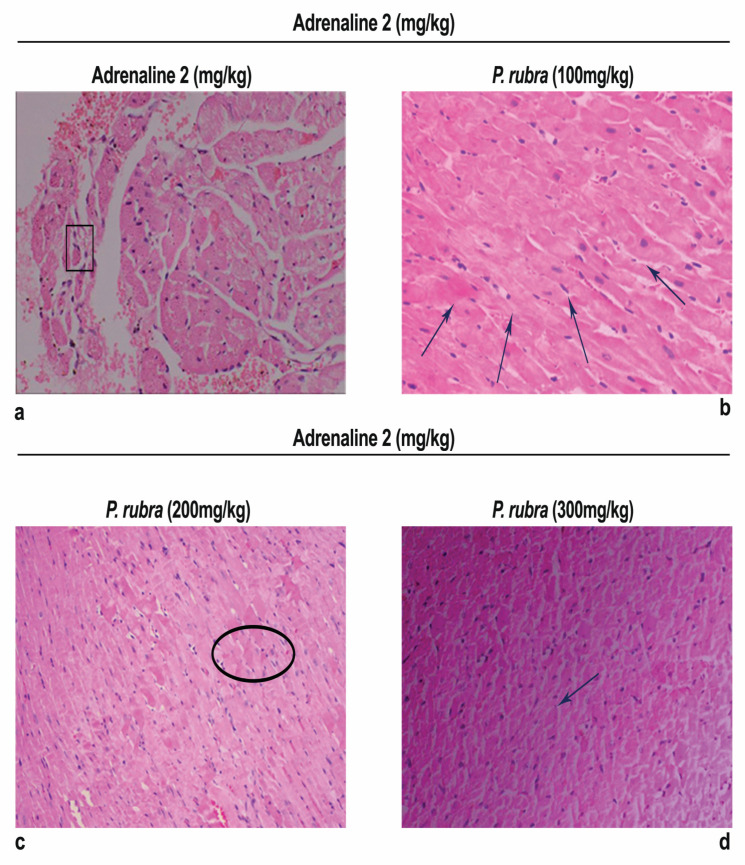
Photomicrograph showing histopathological variations of ventricle section of rabbit heart: (**a**) ADR-intoxicated group; (**b**) *P.rubra* 100 mg/kg + ADR; (**c**) *P. rubra* 200 mg/kg; (**d**) *P. rubra* 300 mg/kg + ADR. In comparison to the ADR-intoxicated group, less inflammatory cells, cardiomyocyte deterioration, infiltration and fibrosis were observed in a dose-dependent manner. Rectangular shape indicates enlarged cardiomyocytes, upward arrows indicate cellular infiltration, oval shape indicates cardiac fibrosis and downward arrows indicate dense normal cluster of cardiomyocytes. Magnification is ×100.

**Figure 7 molecules-27-00251-f007:**
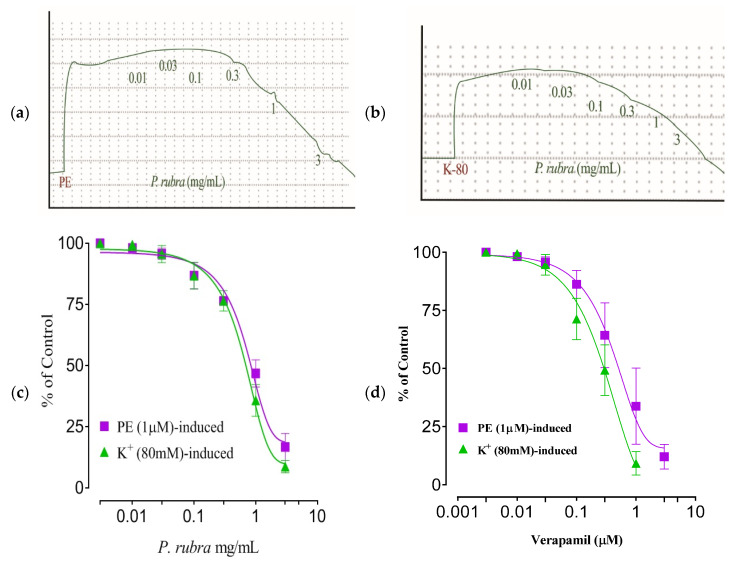
Tracings of aqueous-methanolic leaf extract of *P. rubra* against PE (1 μM)-induced contraction in rabbit aorta strip (**a**); K^+^ (80 mM) induced contraction (**b**); vasorelaxing dose–response curve of *P. rubra* (**c**); and verapamil (**d**) against PE (1 μM) and K^+^ (80 mM)-induced contractions). All the values (*n* = 5) are depicted as mean ± SEM.

**Figure 8 molecules-27-00251-f008:**
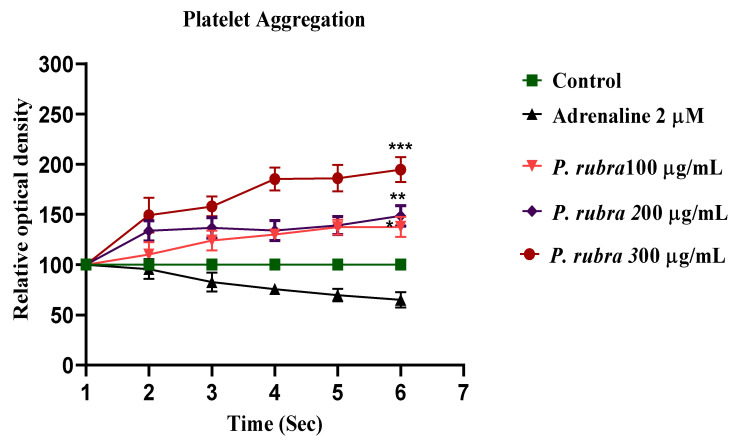
Anti-aggregatory effect of aqueous-methanolic leaf extract of *P. rubra* against ADR-induced aggregation on human platelet. All three doses of *P. rubra* showed significant anti-aggregatory potential in a dose-dependent fashion (*n* = 5).

**Table 1 molecules-27-00251-t001:** Phytochemical investigation of *P. rubra* aqueous-methanolic leaf extract.

Tests	Observation	Results
Alkaloid	PPT	Present
Phenols	Light purple colour	Present
Tannins	Light purple colour	Present
Coumarins	Yellow fluorescence	Present
Saponins	1 cm froth	Present
Anthraquinones	Pink colour	Present
Flavonoid	Light yellow colour	Present

PPT = precipitate formation; froth = presence of froth in test tube.

## Data Availability

Not applicable.

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
