# Peer review of "Pharmacological Justification for the Medicinal Use of *Plumeria rubra* Linn. in Cardiovascular Disorders"

_molecules, 2021, doi:10.3390/molecules27010251_

Round 1
Reviewer 1 Report
This is very good paper proving the therapeutic role of Plumeria rubra (L.), a traditional folkloric medicinal herb used to treat cardiovascular disorders.
I have some minor points.
- Check the format of reference list. Some are not followed journal's policy.
- Figure legends are not well prepared. Please add more information in each figure legend.
- Discuss potential role of individual components in cardiovascular symptoms.
- A lot of typo erros should be fixed (eg., space between numeric and units, etc.
- LC-MS/MS spectrum is required for further characterization of this extract.
Author Response
Dear Sir, kindly consider the revised version of the manuscript. if something left unaddressed kindly suggest. Thanks for your valuable comments and suggestions. All possible changes are made and marked green. thanks and regards

Reviewer 2 Report
The present research aimed to justify by pharmacological studies the therapeutic effect of the plant species Plumeria rubra Linn in cardiovascular disorders. For this, in vitro, ex vivo and in vivo models were used from which it was possible to suggest the possible mechanisms of action associated with the antioxidant, anticoagulant, vasorelaxant and cardioprotective effects of the leaf of the mentioned plant species.
Although the study is interesting, the way in which the subject is approached does not make it attractive enough to the reader since it would seem that the use of the plant is more common in certain regions of the world than in others. Specifically, authors need to focus on improving the following:
- Introduction: Mention the most specific relationship between the mechanism of bioactive compounds and the effects found from the pharmacological evaluation of the different parts of the plant species P. rubra.
- Results: This section is not presented in an adequate way, it is very disorganized, the figures are distorted and some of them have very low resolution or are very small in size. For example, the table presented from the phytochemical evaluation is not clearly described in the text and there are terms in the table that are not understood (e. g. 1cm froth). The HPLC chromatograms are very small and the peaks corresponding to the retention times of the detected compounds are not well distinguished. The micrographs of the histopathological study do not have good resolution and the alterations mentioned in the text are not indicated in the images. The graphs and curves of the preparation of the aortic tissue and the evaluation of the vasorelaxing activity of the aqueous-methanolic extract of the P. rubra leaf did not have good size or resolution. The experiments need to include a positive control. In addition, a good description of the results obtained in the evaluation of acute oral toxicity is not made.
- Discussion: It is necessary to go more deeply into each of the metabolic and antioxidant parameters that were quantified and suggest the possible mechanism by which their values are modified in the presence of the aqueous-methanolic extract of the P. rubra leaf. Furthermore, it is not clear what the term LVH means, because it is not mentioned from the Introduction, but only from the Results. It is also necessary to justify the use of the animal model (rabbit), why weren't rats used, for example?
- The Materials and Methods section is described in a very general and ambiguous way in some sections, as in the case of the determination of the mean lethal dose in the acute oral toxicity study, based on which method was it performed?
- There are many typographical and drafting errors that must be corrected (lack of spaces or extra spaces between words, the name of the plant in some parts of the manuscript is written differently, etc.), for which the manuscript with some of these observations marked in yellow so that the respective corrections can be made. However, it is necessary that the manuscript be reviewed and edited by a native English speaker.

Author Response
Dear Sir, please accept our deepest appreciation for your valuable and detailed suggestion. All possible changes have been made in revised manuscript and changes are marked green. kindly consider the revised MS and welcome your valuable comments in the betterment of manuscript. regards

Round 2
Reviewer 1 Report
Authors did not fully address all the comments. Therefore, this paper is not acceptable.
Author Response
Dear Sir/Madam,
the revised version of manuscript is upload and the the kind suggestions from your side addressed and marked green and red for the editorial suggestions in the manuscript for easy referencing. Dear sir/madam it requested, kindly consider the revised version for the publication as best possibly revised in light of yours, reviewer 2 and respected editor.
kind regards,

Reviewer 2 Report
Although some of the observations were answered, others were not considered and it was expected that a valid justification would be included in each case. For example, why a positive control was not used in the in vivo and ex vivo experiments has not been justified. Table 1 is not adequately described in the text and the meaning of the terms "Ppt" and "1cm froth" has not been specified. Figures 2 and 8 do not indicate the "n" used, nor do they specify whether there is a significant difference between the different concentrations evaluated of the plant species P. rubra with respect to the negative control.
On the other hand, the format of the references must be homogeneous. Also, it is still necessary for the article to be proofread by a native English speaker.
Author Response
Dear Sir/Madam
The revised version of the manuscript is uploaded best possibly revised in light of yours, reviewer 1 and respected editor's suggestions. it is our request kindly consider for the publication.
kind regards,

Round 3
Reviewer 2 Report
Although most of the observations have been answered, it has not been justified why a positive control was not used in the in vivo and ex vivo experiments.
Also, the article still needs to be proofread by a native English speaker.
Author Response
Dear Sir, kindly consider the revised manuscript.
